# On the Possible Trade-Off between Shoot and Root Biomass in Wheat

**DOI:** 10.3390/plants12132513

**Published:** 2023-06-30

**Authors:** Harun Bektas, Christopher E. Hohn, Adam J. Lukaszewski, John Giles Waines

**Affiliations:** 1Department of Agricultural Biotechnology, Siirt University, Siirt 56100, Turkey; 2Department of Botany and Plant Sciences, University of California, Riverside, CA 92521, USA; christopher.hohn@kayagene.com (C.E.H.); adam.lukaszewski@ucr.edu (A.J.L.); waines@ucr.edu (J.G.W.)

**Keywords:** root/shoot ratio, trade-off, biomass allocation, bread wheat, drought stress

## Abstract

Numerous studies have shown that under a limited water supply, a larger root biomass is associated with an increased above-ground biomass. Root biomass, while genetically controlled, is also greatly affected by the environment with varying plasticity levels. In this context, understanding the relationship between the biomass of shoots and roots appears prudent. In this study, we analyze this relationship in a large dataset collected from multiple experiments conducted up to different growth stages in bread wheat (*Triticum aestivum* L.) and its wild relatives. Four bread wheat mapping populations as well as wild and domesticated members of the Triticeae tribe were evaluated for the root and shoot biomass allocation patterns. In the analyzed dataset the root and shoot biomasses were directly related to each other, and to the heading date, and the correlation values increased in proportion to the length of an experiment. On average, 84.1% of the observed variation was explained by a positive correlation between shoot and root biomass. Scatter plots generated from 6353 data points from numerous experiments with different wheats suggest that at some point, further increases in root biomass negatively impact the shoot biomass. Based on these results, a preliminary study with different water availability scenarios and growth conditions was designed with two cultivars, Pavon 76 and Yecora Rojo. The duration of drought and water level significantly affected the root/shoot biomass allocation patterns. However, the responses of the two cultivars were quite different, suggesting that the point of diminishing returns in increasing root biomass may be different for different wheats, reinforcing the need to breed wheats for specific environmental challenges.

## 1. Introduction

From the earliest days of plant domestication, the selection/breeding focus has always been on the above-ground parts of crops [1]. In non-root crops, the selection on the below-ground parts, if any, was only indirect, as a response to selection pressure exerted on the above-ground parts. Only in recent decades have the roots of crop plants gained serious attention as it became clear that in breeding for tolerance of various stresses, and specifically water stress, progress cannot be achieved without a good understanding of the genetics of root systems [2,3,4,5,6]. This resulted in a string of publications on root characteristics and genetic factors that control them in many crop species and model plants [7,8,9,10,11,12,13,14]. Progress has been particularly fast in crops such as maize [15]. Even in wheat, which appears to somewhat lag behind, numerous genome regions that control various aspects of the root system characteristics have been identified, even though not a single gene has been characterized so far [12,16,17,18,19,20,21,22,23,24,25,26,27,28,29], except for the *12-OXOPHYTODIENOATE REDUCTASE* (*OPRIII*) gene [30], which is reported to affect root architecture at different dosages.

Roots absorb water and nutrients, anchor the plant to the soil and also synthesize plant growth hormones. The shoots utilize these resources for photosynthesis and are the site of sexual reproduction. All these functions must work together in coordination for the plant to thrive within its environment. Generally, plants maintain a fairly strict harmony between shoot and root biomass partitioning [31,32]. However, this partitioning fluctuates during different growth and developmental stages. In the early stages of growth, resource allocation and biomass accumulation focus on the roots but that shifts considerably as the plant reaches flowering when a major portion of photosynthates is directed to the shoots [33,34]. Fageria [35] demonstrated that the root-to-shoot mass ratio in wheat, as well as other crops, decreased as plants advanced in age. For these reasons, it appears sensible to study not only the characteristics at some specific points in time, be it young seedlings, heading, or maturity, but also the patterns of changes during development as these may affect the ultimate grain yield in wheat and other crops as well.

It is beyond discussion that root characteristics, such as total biomass [35,36,37], depth [38,39], dispersion in the soil profile [40], and water carrying capacity (which probably means the diameter and the number of vessels) have a direct effect on crop performance [40,41]. This is likely due to the ability of a larger root system to absorb more water and nutrients from the soil; an added benefit is reduced leaching and agricultural run-off [42]. Root systems widely dispersed in the soil profile close to the surface are a good adaptation to irrigation [43]. On the other hand, roots growing straight down into deep soil layers would be able to reach water not available to shallow roots [44,45,46,47,48,49] and may be an adaptation to rain-fed conditions. It is expected that with a better understanding of root traits and increasing knowledge of root genetics, specific root ideotypes can be devised and applied in cultivar development. 

The root system is only a part of an organism and so it appears plausible that additional investment into an extensive root system may affect the above-ground parts of the plant [35,40,50,51]. This aspect appears worthy of a detailed study, as it may point to traits of the root system that would be beneficial in a given environment and how they might impact the grain yield if conditions change. Perhaps a large root system is beneficial when water is limited; however, will it remain an advantage or become a burden when water becomes sufficient? These types of questions need to be answered before efforts are made toward modifying crop root system traits. In this study, we examined a large body of results from numerous studies of root systems in wheat, grown and tested under various conditions of water availability, in different experimental systems, at different stages of development, and in a wide range of genetic resources, for signs of a possible trade-off between the below-ground and above-ground parts of plants and we demonstrate that such a trade-off exists. We then performed a preliminary study of such trade-offs in a system with different water availability to plants and confirm that they exist but may be different in different wheats.

## 2. Results

### 2.1. Combined Data Analysis

The main objective here was to perform a meta-analysis of combined data from a series of previous experiments. Those experiments, conducted with diverse sets of wild and domesticated accessions from the Triticeae tribe, offered a wide window of root and shoot biomass variation measured between 21 days after planting to maturity. A total of 6353 data points were assembled to create the scatter plots for the relationship between the root and shoot biomass (Figure 1). The data from the previous experiments that were continued for 21–28 days, 40 to 70 days, and until maturity were analyzed as separate groups to obtain ranges in SM and RM. The first set included 1243 data points (Figure 1a) from experiments where plants were grown to maturity. The second and third sets had 1342 (40–70 days) and 3768 points (21–28 days) (Figure 1b,c), respectively. These experiments were concluded before maturity, with only the shoot and root biomass collected (Table 1). As previously reported in our publications [13,14,52,53,54,55,56], each set had significant genetic diversity for all evaluated traits. Here, two traits of interest are given in a summary table (Table 2). Shoot biomass for 21–28 days, 40–70 days, and maturity experiments ranged between 0.05–3.00 g, 1.54–70.19 g, and 44.07–117.11 g, respectively. Similarly, the root biomass for the same experiments ranged between 0.01–2.29 g, 0.27–12.49 g, and 3.96–17.68 g, respectively. Overall, there was a strong correlation between the root and shoot biomass in all experiments. Each experiment was conducted until a certain phenological stage, e.g., booting and anthesis. A total of 6353 data points from the entire set of accessions (639 accessions including checks) were used for the correlation analyses and similar trendlines were obtained in all (Figure 1a–c). All three data sets clearly demonstrate a proportional increase in root and shoot biomass early on, and a gradual reduction in the shoot biomass increases (including grain yield, Figure 1a) as the root biomass increases. In each scenario, there appears to be a point where additional increases in root biomass negatively affect shoot biomass. Following this trendline, a set of preliminary validation experiments are designed to observe drought-related changes and trade-offs in root–shoot biomass allocation patterns.

### 2.2. Trade-Off Pot Experiments

Results from the two trade-off experiments were not significantly different and were combined. In pot experiments, Pavon 76 did not show significant differences between drought treatments and the control for days to booting, days to heading, and days to anthesis, with means of 49.1, 54.3, and 59.1 days, respectively. However, days to maturity showed significant differences between the treatments and the control, with means of 92.8 and 137.5 days, respectively. The total number of tillers was also significantly different between the treatments and the control, with means of 9.3 and 34.3, respectively. The total number of fertile tillers showed significant differences within treatments as well as between treatments and the control. Plants experiencing drought-at-booting had a mean number of 5.5 fertile tillers, those with drought-at-heading and drought-at-anthesis had 8.3 fertile tillers, and the control had 32.2 fertile tillers per plant (Figure 2a). Drought treatments significantly reduced shoot biomass to 11.9 g versus 36.0 g for controls. The root biomass was also significantly affected by the treatments. The means for plants receiving drought at booting, heading, and anthesis were 3.5, 4.9, and 5.1 g, respectively; significantly less than the control at 7.1 g. Grain yields for the drought treatments were significantly lower than for the control, which had a mean grain yield of 51.3 g. Plants receiving drought-at-heading yielded the second highest with a mean of 12.0 g and both the drought-at-booting and drought-at-anthesis treatments yielded the lowest with a mean of 6.3 g (Figure 2b).

Yecora Rojo showed a similar pattern to that of Pavon 76: there were no significant differences between drought treatments and the control for days to booting, to heading, or to anthesis with means of 29.6, 36.7, and 41.6 days, respectively, and there was a significant difference for days to maturity between treatments and the control, with means of 77.9 vs. 106.5 days, respectively. The total number of tillers for the drought treatments at 6.6 was significantly lower than the control (9.5). There was a significant difference between the drought-at-booting and drought-at-anthesis for the number of fertile tillers, at 5.0 and 7.0, respectively. Plants receiving drought-at-heading were intermediate with a mean of 6.0, which was not significantly different from the other treatments (Figure 2c). All treatments had lower means than the control with 8.8 fertile tillers per plant. All drought treatments significantly reduced the shoot biomass (mean of 3.1 g) relative to the control (4.7 g). The control had the highest root biomass per plant (2.6 g) and the drought-at-anthesis treatment was the second largest, with 1.9 g per plant. Both the booting and heading drought treatments had the same mean of 1.3 g per plant. Grain yield also showed a significant difference within the treatments and between the treatments and the control. The control yielded 11.3 g grain per plant, the drought-at-anthesis yielded 5.7 g per plant, and the drought-at-heading treatment yielded 4.7 g per plant, which were not significantly different from the drought-at-anthesis treatment or the drought-at-booting treatment, and the drought-at-booting treatment yielded the lowest with 3.6 g per plant (Figure 2d).

### 2.3. Trade-Off Tube Experiments

In the tube experiments, Pavon 76 did not show significant differences between the treatments and control for days to booting, days to heading, days to anthesis, or days to maturity with means of 55.1, 60.6, 64.8, and 125.2, respectively. The number of tillers was significantly different within treatments and between the treatments and control. The shallow treatment had the largest number of tillers per plant (28.5), the control was second with 18.3, and the deep treatment had a mean of 5.0 tillers per plant. Of the total number of tillers, 27.3, 16.3, and 4.0 were fertile for the shallow, control, and deep treatments, respectively, with significant differences among all of them (Figure 3a). The shoot biomass also varied significantly among the treatments and the control, with means of 38.2, 20.1, and 5.7 g for the shallow, control, and deep treatment, respectively. Root biomass above 30 cm followed the same trend with means of 6.3, 2.7, and 0.9 g for the shallow, control, and deep treatment, respectively. Root biomass below 30 cm showed no significant difference between the treatments and the control with a mean of 2.1 g (Figure 3b). However, the total root biomass showed significant differences within treatments and between treatments and the control. In the control and the deep treatment, roots reached the bottom of the 1 m tube and, as expected, roots in the shallow treatment did not grow much into the anaerobic volume of sand saturated with water below 50 cm with a mean length of 57.3 cm. Both treatments and the control varied significantly for grain yield with means of 54.1, 29.5, and 7.5 g per plant for the shallow, control, and deep treatments, respectively.

Yecora Rojo did not show significant differences in the tube experiments for days to booting, days to heading, or days to anthesis with means of 36.5, 42.4, and 46.6 days, respectively. For days to maturity, there was a significant difference within treatments and between the deep treatment and the control. The deep treatment had a mean of 79.0 days to maturity while the control and shallow treatment had a mean of 116.1 days. The total number of tillers was significantly different for the control and deep treatment which had 10.8 and 2.0 tillers per plant, respectively. The shallow treatment was intermediate and not significantly different from either of the other two, with a mean of 6.5 tillers. The number of fertile tillers was not significantly different between the control and shallow treatment with 9.3 and 6.0 fertile tillers per plant, respectively; the deep treatment had a mean of 2.0 fertile tillers per plant (Figure 3c). The shoot biomass of the two treatments was not significantly different due to large variances of the groups with means of 3.2 and 0.76 g for the shallow and deep treatment, respectively. Both treatments were significantly different from the control which had a mean shoot biomass of 7.3 g per plant. Root biomass above 30 cm was significantly different within treatments and between treatments and the control, with means of 1.8, 1.1, and 0.2 g per plant. Root biomass between the treatments and the control was significantly different but the treatments has similar mean biomass of 0.3 g per plant. The control had a mean of 4.5 gr per plant. The total root biomass did not differ between the shallow and deep treatments (1.2 and 0.7 g per plant), while the control was significantly higher, with 6.3 g of total root biomass per plant (Figure 3d). In the control and the deep treatment, roots reached the bottoms of the 1 m tubes while the shallow treatment had a mean root length of 45 cm. Grain yield was significantly different between the treatments and between treatments and the control. The control yielded a mean of 11 g grain per plant, the shallow treatment yielded 6.1 g, and the deep treatment yielded 2.1 g per plant.

Cvs. Pavon 76 and Yecora Rojo were significantly different for all traits except the root length and the root mass below 30 cm. The contrasts between cultivars in biomass were so large that results had to be displayed on graphs with different scales (Figure 3 and Figure 4). Scatter plots for root biomass plotted against shoot biomass and grain yield for cvs. Pavon 76 and Yecora Rojo are shown in Figure 4. These data were combined from both systems used in the trade-off experiments.

## 3. Discussion

The root and shoot carbon allocation patterns are not fully understood [58], but the differences, expressed as ratios of the root-to-shoot biomass may decide the plant’s fate, survival, and fitness [59]. Due to the plasticity of roots, environmental variables, diverse responses of species/genotypes to external factors, and a limited time window for each study, it is not yet possible to draw clear-cut conclusions [60,61]. In this study, we attempted to evaluate the root–shoot relationships under various water availability scenarios using a large set of data from diverse sets of wheat accessions and a preliminary experiment to validate the conclusions.

### 3.1. Genetic Variation for Root Biomass

In general, root and shoot biomass were directly related to heading dates. As the experiments were run for longer durations the correlation values between the shoot and root biomasses increased. In the experiments described here, an average of 84.1% of the variation seen in mapping populations Sonora × Foisy (SF), Sonora × CBDeM (SC), and CBDeM × Foisy (CF) was explained by a positive correlation between shoot and root biomass, and the heading date explained an average of 88.8 and 78.1% of the variation for those traits, respectively. It was obvious that in these populations’ loci controlling the heading date had a major effect on the shoot and root biomass. The longer a plant grows (or, perhaps, more slowly), the larger it becomes, and the growth of the shoot and root appear to be balanced. There is an apparent correlation between the extended vegetative growth for the above and below-ground parts [58,62]. However, the question of which trait (root or shoot) drives the other, and a generalized model for the peak of the curve for root/shoot ratio is still a mystery. To untangle the relationships among heading date, shoot biomass, and root biomass, populations or accessions with similar phenology should be used [25].

### 3.2. Trade-Offs in Root and Shoot Biomass

The combined data, comprising 6353 data points from a range of wild and cultivated wheats, indicate that in the initial stages of growth, the shoot and root biomasses increase proportionately; however, at a certain point, as the rate of the root biomass continues to increase, the increase in the shoot biomass begins to slow down (Figure 1). The sharpest divergence between the shoot and root biomass was between the 40th and 70th days of growth (Figure 1b). On the other hand, when all data were plotted for the root-shoot biomass trends, the decline in shoot biomass was less obvious, suggesting an equilibrium between above and below-ground biomass allocation (Figure 1a). This almost linear interaction between the above- and below-ground biomass distribution under optimal growing conditions (full water availability) is dramatically affected when water stress is induced (Figure 4). To analyze this interaction more closely, we tested two cultivars in two experimental systems (here pots and tubes) and under various water availability regimes designed specifically to differentially affect either the root or the shoot growth. The two cultivars had been extensively tested before and are known for drastic contrasts in terms of root and shoot biomass growth, as well as in phenological traits such as heading dates and plant height [36,63,64,65]. 

Pavon 76 and Yecora Rojo have similar shoot and root biomass distributions, but different relationships between the root biomass and grain yield (Figure 4). Under optimal conditions, there is an almost linear correlation between the root and shoot biomass in both cultivars. This relationship is consistent with the general pattern provided by the meta-analysis since the two are located at opposite ends of the trend line. In Pavon 76, the root biomass and grain yield increase in parallel; in Yecora Rojo, as the root biomass increases, the grain yield drops. Yecora Rojo closely follows the general pattern apparent from the combined data; Pavon 76 appears as an outlier. These two examples imply that while a general relationship exists between the root and shoot biomass, and those two and the grain yield, individual lines or cultivars may substantially deviate from it. A notable feature of Pavon 76 is its ability to maintain relatively higher root biomass under water stress conditions [66]; while in Yecora Rojo the rate of root biomass growth drops as the stress level increases (Figure 2 and Figure 3). This difference may be explained by different carbon allocation patterns; in some cultivars, the carbon sinks are altered more dramatically by environmental factors than in others. In this case, while Pavon 76 shows a positive interaction between the root biomass and grain yield, it is quite distinct from the common behavior of other genotypes. 

This issue seems to be worth further study; a dramatic difference in the water stress response between two cultivars originating from the same breeding program (CIMMYT: International Maize and Wheat Improvement Center) and selected under very similar conditions (surface irrigation) suggests that the root–shoot–grain yield relationships are complex. Watt et al. [6] highlighted the importance of the root–shoot relationships when carbon stocks are limited. The effect of root carbon allocation on shoot biomass or grain yield becomes more dramatic when plants are under stress conditions. The root/shoot biomass ratios for Pavon 76 and Yecora Rojo (Table 3) highlight the changes in the carbon allocation patterns under drought stress. In both cultivars, when drought stress is induced, the total root and shoot biomass is reduced. Drought appears to change carbon sinks in favor of the root, and the root: shoot ratio increases. This shift is apparent in both cultivars but Pavon 76 seems to use its wide plastic response-ability to maintain a higher shoot growth rate [66]. When an ample amount of water is provided, the root and shoot biomass produced was large. In fact, the root/shoot ratio was the lowest when an excessive amount of water was available at a relatively shallow depth of 50 cm. Ample water at this level apparently reduced or eliminated the need for deep root growth, but the total root biomass was the highest among all treatments (Table 3, Figure 3) and this translated into a larger shoot biomass. On the other hand, when roots were forced to chase the ever-dropping water level, plants of both cultivars produced the lowest root and shoot biomasses, but the highest root/shoot ratios. The peaks of root–shoot balance could perhaps reflect the changes in allocation, by redirecting resources to above ground. But water chasing experiments (deep treatment) showed that the proportion of shoot/root was inverted. So, under certain conditions, such as water-chasing conditions here, more resources are allocated to roots, at the expense of shoots.

In a different stress scenario, in the pot experiment, the water stress was induced at booting, heading, or anthesis, and there were no means of escape from it, such as in the “deep treatment” in a tube. The timing of the stress onset had a clear effect on the plant’s ability to cope with it. The later the onset of terminal drought, the higher the biomass and grain yield. The correlation between the above- and below-ground biomass is still evident. Even with Yecora Rojo and Pavon 76 being so different in the plasticity levels and growth rate, they both followed a similar trend for biomass allocation, and the trend to a higher root ratio under stress was the common behavior. 

Understanding the cultivar/genotype-specific biomass allocation and grain yield relationships may provide a better understanding of stress adaptation and resource management [58,67]. Cultivars with extended rooting ability may not always be advantageous under drought conditions [51]. The relative water use efficiency may dramatically limit grain yield under stress. Figueroa-Bustos et al. [68] observed a sharper decline in the grain yield and the thousand kernel weight under terminal drought in a cultivar with a large root system (Bahatans-87) than in a cultivar with smaller roots (Tincurrin). They suggested that the large vegetative development of Bahatans-87 depleted the available water more quickly. So, an effective root system distributed deep in the soil may be more efficient under terminal drought. However, the root system size may not solely solve the puzzle. Figueroa-Bustos et al. [69], Figueroa-Bustos et al. [70], and Bektas et al. [54] reported extensive rooting ability in cultivars with longer vegetative growth, which may be useful under drought occurring at later growth stages, such as anthesis, but may also limit the plant’s ability to direct adequate resources to fill the grain. Therefore, not only the rooting potential but also the relative root growth dynamics and size become important. Van der Bom et al. [58] highlight the advantage–disadvantages of various root ideotypes depending on environmental fluctuations. A narrow, small, and deep “steep, cheap, and deep” root system [67] may be advantageous under one set of conditions but may limit grain yield under another [51]. Plants actively explore the soil profile and even sense neighboring plants, and there is a balance between an increased root surface area for resource capture and above-ground photosynthetic ability. Under limited carbon storage, each genotype appears to have some specific behavior, while under optimum resource availability, some genotypes may overproduce roots risking their future if conditions change (Table 1; Pavon 76 under excessive water availability). Cabal et al. [71] suggested an evolutionary stable strategy (ESS) as a naturally evolved behavior to cope with neighbors and environmental fluctuations. Therefore, we can correlatively speculate a genotype-level response strategy to water and soil resource fluctuations. Is there a common cost–benefit algorithm, or do genotypes individually decide their fate? The observations presented here appear to support the individual algorithm theory. While both cultivars directed resources to root development under water limitation, their resource allocation patterns/rates were different. The results of the root and shoot biomass data from the large dataset provide a broad conclusion about the complex carbon traffic between the above and below-ground plant parts, but that cannot guarantee a generalized model at the genotype level.

## 4. Materials and Methods

### 4.1. Plant Material

The analyses presented here combine data from multiple experiments run over several years at the University of California, Riverside, with a wide range of wheat cultivars, genetic stocks, and wild relatives of wheat (Table 1). These included three mapping populations consisting of doubled haploids generated from hybrids of CVs. Sonora, Foisy, and Chiddam Blanc de Mars (CBDeM) described in Hohn and Bektas [57]. Cv. Foisy was chosen by Mr. Foisy in Oregon in 1865, CBDeM originates from Ville de Paris, France, and was chosen from an English landrace. Sonora was a landrace in Durango, Mexico, known for good drought tolerance [55]. The fourth mapping population, SynOpDH, was a cross of Synthetic W7984 (a synthetic amphiploid developed from the durum wheat line ‘Altar 84’ (*Triticum turgidum* L.), combined with the accession (219) ‘CIGM86.940’ of *Ae. tauschii*), and Opata M85, a well-known CIMMYT bread wheat cultivar [54]. The other experiments included a variety of materials, including wild and domesticated Triticeae species with diploid, tetraploid, and hexaploid ploidy levels, Turkish landraces and cultivars, CIMMYT wheats (historical cultivars), data from allelic variation experiments with various 1RS.1BL translocation lines [55,56], and from wild and domesticated Triticeae and Aegilops sp. experiments [13,14,52,53]. 

The final set of data consisted of the trade-off validation experiments conducted with two semi-dwarf spring wheat cultivars from CIMMYT, Pavon 76, and Yecora Rojo (Table 1).

### 4.2. Experimental Design for Combined Data

Raw data sets from experiments running for a similar duration were combined to fit into similar scales. Following these criteria, three sets of data were created based on experimental duration. A summary of the experimental design for each set and resources for the collected data is given in Table 1. All data used and generated in this study are from experiments conducted at the University of California, Riverside, and data used for the combined data analyses have been published [13,14,52,53,54,55,56,57]. Generally, two systems were used. One consisted of PVC tubes (80 or 100 cm in length) each fitted with a plastic sleeve filled with silica sand with a bulk density of 1.42 g mL^−1^ and 24% field water capacity (*w*/*w*). In the standard system, water was applied to the top of the tube [72]. The second system consisted of 3.8 l pots each lined with a plastic sleeve with holes punctured at the bottom for drainage, and filled with sand. Pots were brought to the water holding capacity and allowed to drain for 24 h before planting. Seeds were imbibed for 24 h and planted. In all experiments, Peters Excel fertilizer (21-5-20 N-P-K, www.scottspro.com (accessed on 20 May 2023)) was injected into the irrigation water at the 1:100 ratio. Plants were irrigated with ample amounts of water to keep the sand at the water-holding capacity, equally for each tube/pot. Plants were grown under natural photoperiod in a temperature-controlled glasshouse (20–30 °C and 50–90% relative humidity).

### 4.3. Experimental Design for Trade-off Study

The system devised for the trade-off study consisted of the same PVC tubes as above, but with a much longer plastic sleeve lining extending beyond the bottom of the tube and bent upwards in a U-shape (Figure 5). Water was delivered through that extended sleeve, hence from the bottom up in the tube with sand, and the water level was closely monitored to always be below the maximum reach of the roots, while the capillary upward movement of water in the sand was minimal.

After the experiments were terminated, shoots were separated from the roots, washed clean of sand, dried for 72 h in a forced air drier at 80 °C, and weighed for the total dry biomass, as described in Bektas [52] and Hohn [55]. During the experiment, days to booting, heading, anthesis, and maturity were recorded. For the experiments completed at maturity, the heads were harvested, and the shoots were cut at the soil surface to separate them from the roots. Shoot biomass is reported without grain yield. Heads were threshed to record the grain weight for each plant. For tube experiments, root lengths were measured and root biomass above 30 cm (shallow roots) was separated from the root biomass below 30 cm (deep roots) and weighed separately.

### 4.4. Growth Conditions and Irrigation Regimes on Trade-off Experiments

Cvs. Pavon 76 and Yecora Rojo have been used extensively as standards in root studies conducted at the University of California, Riverside by Dr. J. Giles Waines and his co-workers. With a substantial body of performance data, these cultivars were chosen for preliminary experiments aimed at observing the relationships between the shoot and root growth. 

In tube-based experiments, two treatments plus a control were run for each cultivar in a factorial design with two replications. The control was given 500 mL of water daily, from the top down. The first treatment received water from the bottom only (deep treatment) starting at two weeks of growth. Water from the bottom was kept at the furthest point at which the roots reached within the sand as visible through the clear plastic sleeve of two check tubes. As the roots grew into the water profile the water level continuously dropped until the roots reached the bottom of the tube. The second treatment received water from both the top and the bottom daily (shallow treatment). Water from the bottom was maintained at 50 cm and 500 mL of water was added each day from the top. 

Pot experiments were carried out in three replications set up in a factorial design with three treatments and a control. Each pot was kept at the water holding capacity until three different phenological stages: booting, heading, and anthesis; at these points, water was cut off entirely. Treatments were termed drought-at-booting, drought-at-heading, and drought-at-anthesis. After that point, any plant showing severe water stress was given water to prevent death. This point was determined when leaves began to wilt and curl beyond mild symptoms. The control was given ample water to maintain the sand at the water-holding capacity throughout the experiment.

### 4.5. Statistical Analysis

The analyses of variance (ANOVA) for traits in each experiment were based on the mean values of the experimental units and were considered significant at *p* ≤ 0.05. Genotype means were used for the Least Significant Difference (LSD) all-pairwise comparisons where α ≤ 0.05. Pearson’s correlation coefficients for the shoot and root biomass were calculated on a mean basis across replications [73]. In total, raw values for 6353 individuals were used to create scatter plots fitted with a Loess smoothing curve with an alpha of 0.75 with a quadratic degree using statistical analysis software, Statistix 10 (Analytical Software; Tallahassee, FL, USA).

## 5. Conclusions

It is well known that traits such as root biomass and root-to-shoot ratios are genetically and environmentally controlled. The results presented here make it obvious that these traits are highly complex, and, in many cases, environmental effects are so high that drawing out differences between genotypes becomes impossible. However, the accumulated data permit some generalizations on the root-and-shoot relationship in wheat. Generally, the shoot and root biomasses increase in parallel, but only up to a certain point. From that point on, additional increases in root biomass negatively impact increases in above-ground biomass and/or grain yield. This is possibly due to an imbalance of resource allocation and the high cost of maintaining a large root system. The position of the point of change is different for different genotypes so these generalizations cannot be blindly applied to all genotypes.

## Figures and Tables

**Figure 1 plants-12-02513-f001:**
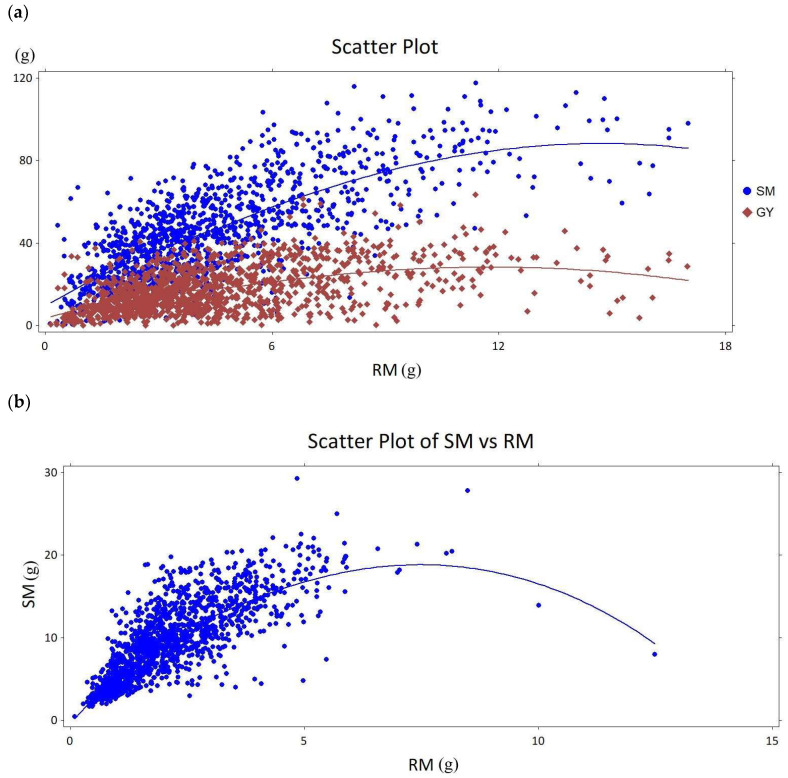
Scatter plots for the combined data of root biomass (RM; g) and shoot biomass (SM; g), and grain yield (GY; g) in various experiments conducted at the University of California, Riverside. Each point represents a single plant from experiments carried to; (**a**) maturity, (**b**) for 40 to 70 days, and (**c**) for 21 to 28 days. Plot a shows the relationship between the root mass and the shoot mass or grain yield (GY); plots b and c show the relationships between the root mass (RM) and the shoot mass (SM). Note that plots are on different scales (Loess α = 0.75).

**Figure 2 plants-12-02513-f002:**
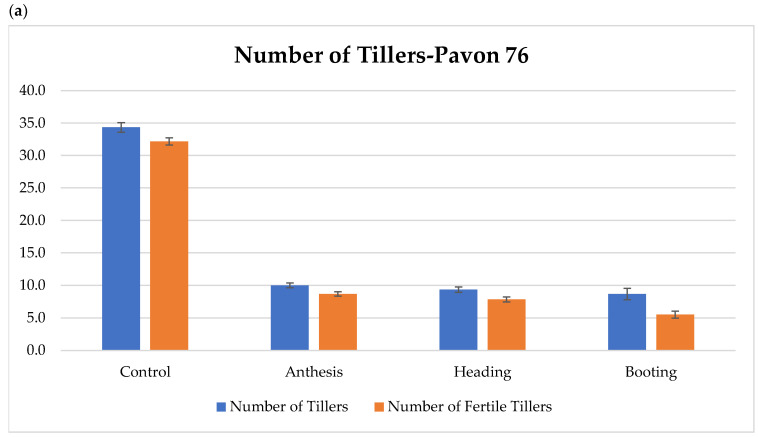
Bar graphs displaying the results for (**a**,**b**) Pavon 76 and (**c**,**d**) Yecora Rojo from the pot system used in the trade-off experiments. Means followed by different letters within columns are different according to the least significant difference (LSD) test at *p* ≤ 0.05. Letter designations: a–c: grain yield (Yield), d–e: shoot biomass, f–h: root biomass. Treatments were: drought at anthesis, drought at heading, and drought at booting.

**Figure 3 plants-12-02513-f003:**
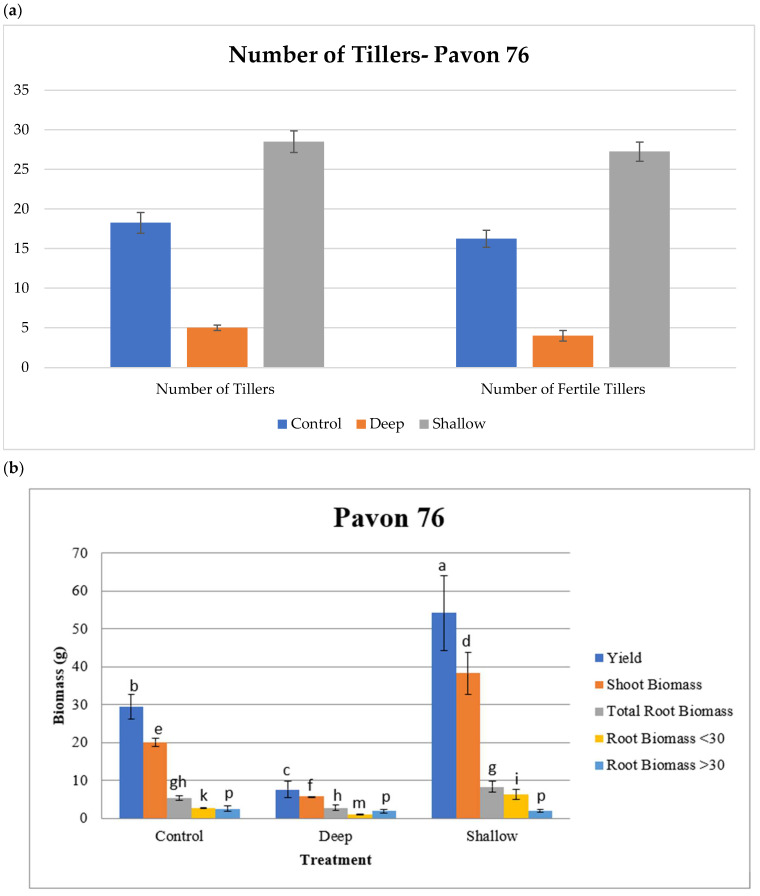
Bar graphs displaying the results for (**a**,**b**) Pavon 76 and (**c**,**d**) Yecora Rojo from the tube system used in the trade-off experiments. Means followed by different letters within columns are different according to the least significant difference (LSD) test at *p* ≤ 0.05. Letters are designated alphabetically (each trait given with different letter groups) and note that the groupings were different for Pavon 76 and Yecora Rojo. Yield: grain yield.

**Figure 4 plants-12-02513-f004:**
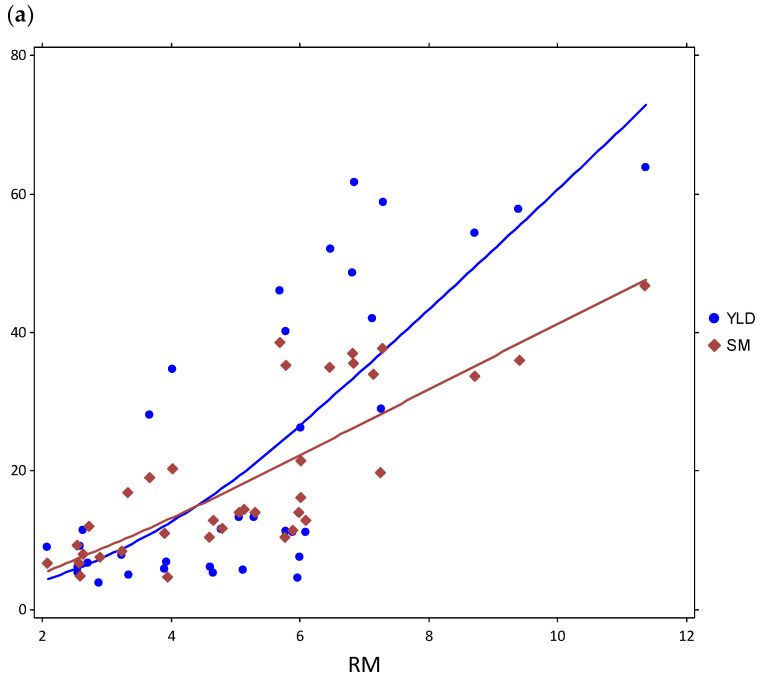
Scatter plots for root biomass (RM) in grams plotted against shoot biomass (SM) and grain yield (YLD) in grams for cultivars (**a**) Pavon 76 and (**b**) Yecora Rojo. Data shown are combined from both systems used in the trade-off experiments. Note that scatter plots are on different scales (Loess α = 2.0).

**Figure 5 plants-12-02513-f005:**
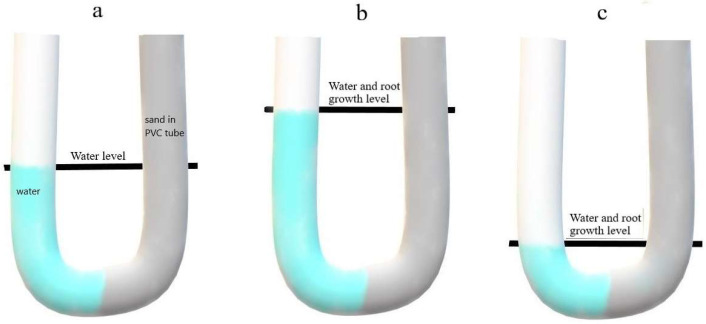
Illustration of the trade-off PVC tube experiment conducted with cvs. Pavon 76 and Yecora Rojo. Shallow treatment (**a**) had an ample amount of water kept at a 50 cm level until the end of the experiment. In deep treatments (**b**,**c**) water level was progressively reduced (early growth stages as (**b**), later as (**c**)) as roots grow deep into the tube, to make roots chase the water. Control tubes were watered daily from the top.

**Table 1 plants-12-02513-t001:** Lists of populations, numbers of accessions, experimental designs, and experiment durations providing data points for analyses here, and references for published results.

	Mapping Populations				Data from Diverse Sets of Lines				Trade-Off Experiments
Population Name	Sonora × Foisy (SF)	Sonora × CBdeM (SC)	Chiddam Blanc de Mars (CBdeM) × Foisy (CF)	Synthetic W7984 × Opata M85 (SynOpDH)	Wheat Wild Relatives	Historical CIMMYT Accessions	Turkish Wheat Accessions	1RS.1BL Translocation and 1B Substitution Lines	Pavon 76 and Yecora Rojo
Number of Accessions evaluated	141	146	128	147	15	9	19	32	2
Growth conditions	PVC tubes and 3.8 l pots			PVC tubes				PVC tubes	PVC tubes and 3.8 l pots
Growth duration	21–28 days in 2014–2015; 30–60 days in 2013; 40 days in 2016			Maturity in 2012–2014				40–70 days between 2012–2015	Maturity in 2015–2016
Reference	[55,57]			[13,14,52,53,54]				[55,56]	

**Table 2 plants-12-02513-t002:** Range of root (RM) and shoot (SM) biomass values for accessions including four double haploid mapping populations (SC, SF, CF, and SynOpDH), and diverse sets of various domesticated and wild accessions (CIMMYT wheats, Turkish wheats, Pavon translocation and substitution lines and wheat wild relatives) evaluated on different growth conditions and durations.

Trait	Duration/Growth Stage	N	Mean (g)	SD	SE Mean	C.V.	Minimum (g)	Median (g)	Maximum (g)
Shoot Biomass	21–28	3768	0.81	0.51	0.01	62.47	0.05	0.66	3.00
	40–70	1342	10.98	9.68	0.27	88.19	1.54	8.78	70.19
	Maturity	1243	46.33	22.43	0.61	48.41	0.36	44.07	117.11
Root Biomass	21–28	3768	0.30	0.22	0.00	73.50	0.01	0.26	2.29
	40–70	1342	2.14	1.27	0.04	59.38	0.27	1.88	12.49
	Maturity	1243	4.76	2.98	0.08	62.54	0.15	3.96	17.68

**Table 3 plants-12-02513-t003:** Root/shoot biomass interactions under various drought scenarios and different irrigation regimes for wheat cvs. Pavon 76 and Yecora Rojo. The tube study had three irrigation regimes; water given from the bottom up and roots chasing water (“deep treatment”), water kept at a constant level of 50 cm (”shallow treatment”), and controls with ample water provided from the top. The pot study had three different drought treatments as drought-at-booting, drought-at-heading, and drought-at-anthesis plus control with optimum irrigation. RM, root biomass; SM, shoot biomass; RM < 30, root biomass below 30 cm; RM > 30, root biomass above 30 cm in the tube system. RM/SM, root biomass/shoot biomass ratio.

Tubes					Pots	
Pavon 76	**Treatment**	**RM/SM**	**RM < 30/RM > 30**	**RM < 30/SM**	**Treatment**	**RM/SM**
	Control	0.26	1.11	0.14	Control	0.20
	Deep	0.49	0.50	0.16	Anthesis	0.40
	Shallow	0.22	3.15	0.16	Heading	0.43
					Booting	0.30
Yecora Rojo	**Treatment**	**RM/SM**	**RM < 30/RM > 30**	**RM < 30/SM**	**Treatment**	**RM/SM**
	Control	0.86	0.41	0.25	Control	0.55
	Deep	0.87	0.54	0.30	Anthesis	0.55
	Shallow	0.39	8.54	0.35	Heading	0.44
					Booting	0.45

## Data Availability

The data presented in this study are available on request from the corresponding author.

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
