# Peer review of "On the Possible Trade-Off between Shoot and Root Biomass in Wheat"

_plants, 2023, doi:10.3390/plants12132513_

Round 1
Reviewer 1 Report
The manuscript entitled “On the possible trade-off between shoot and root biomass in wheat” by Bektas, et al. aimed to understand the relationship between the biomass of shoots and roots in bread wheat (Triticum aestivum) and its wild type. It shows that there is correlation between the root and shoot biomass allocation patterns in the bread wheat. This study provides the preliminary results with different water availability scenarios and growth conditions for two cultivars, indicating that the root biomass of bread wheats vary with many factors, especially the environmental challenges.
Overall, the methods used and the description in the study are thorough. Statistical analysis is provided with the data analysis. The authors clearly describe the observation and conclusions. There are some minor concerns before I recommend accepting it:
-
Line 97 - 99, The authors presented 6353 data points ( 1243 data from maturity, 1342 data from 40-70 days, 3768 data from 21-28 days). Based on table 1, it seems that most of the data were from different experiments. That means, there was no correlation between points from 21 - 28 days and those from 40-70 days. Among 6353 points, are there any points from the same population? Did the author collect the data from the same subjects from 21- 28 days, 40-70 days, until maturity.
-
Line 118, figure 1 legend: it should be “the combined data of root biomass (RM) and ….”. Please double check the text.
-
Line 123 - 126, In figure 1, there was no description about the x-axis and y-axis. Please add the units for biomass in the figure.
-
Line 140 - 145, it is highly recommended that the author should visualize the results between the control and drought treatment, i.e. using bar graphs.
-
Line 186 - 192, Similarly, it is recommended that the author visualize the results for trade-off Tube experiment.
Author Response
Responses to Reviewer 1
Q1: The manuscript entitled “On the possible trade-off between shoot and root biomass in wheat” by Bektas, et al. aimed to understand the relationship between the biomass of shoots and roots in bread wheat (Triticum aestivum) and its wild type. It shows that there is correlation between the root and shoot biomass allocation patterns in the bread wheat. This study provides the preliminary results with different water availability scenarios and growth conditions for two cultivars, indicating that the root biomass of bread wheats vary with many factors, especially the environmental challenges.
Q2: Overall, the methods used and the description in the study are thorough. Statistical analysis is provided with the data analysis. The authors clearly describe the observation and conclusions. There are some minor concerns before I recommend accepting it:
Q3: Line 97 - 99, The authors presented 6353 data points ( 1243 data from maturity, 1342 data from 40-70 days, 3768 data from 21-28 days). Based on table 1, it seems that most of the data were from different experiments. That means, there was no correlation between points from 21 - 28 days and those from 40-70 days. Among 6353 points, are there any points from the same population? Did the author collect the data from the same subjects from 21- 28 days, 40-70 days, until maturity.
A1-3: Thank you for the comments. Yes, data come from a series of experiments conducted over several years/several growing seasons, see Table 1. The experiments are descibed in great detail in original papers; here we only give a fairly general description. Details of each experiment are in previously published articles, items #52 through #57 in references here.
Q4: Line 118, figure 1 legend: it should be “the combined data of root biomass (RM) and ….”. Please double check the text.
A4: Thank you, corrected.
Q5: Line 123 - 126, In figure 1, there was no description about the x-axis and y-axis. Please add the units for biomass in the figure.
A5: We added x and y-axis values in Figures 1a, b and c.
Q6: Line 140 - 145, it is highly recommended that the author should visualize the results between the control and drought treatment, i.e. using bar graphs.
Q7: Line 186 - 192, Similarly, it is recommended that the author visualize the results for trade-off Tube experiment.
A6, 7: Good idea! Thank you. Added on both.
Reviewer 2 Report
I have read the article titled “On the possible trade-off between shoot and root biomass in wheat”, although, the theme of the article is interesting and relevant to the journal’s scope. But writing style and presentation is inadequate and inappropriate. However, I feel numerous flaws in all parts of the paper. Tile is not attractive, change it with attractive one. Improve the quality of fig 2 and 3.
The English language is not of standard quality which needs substantial improvement. The writing style is informal. Many sentences have no structure, keep consistency in writing. English language needs to be attested by a qualified English language specialist.
Author Response
Responses to Reviewer 2
Q1: I have read the article titled “On the possible trade-off between shoot and root biomass in wheat”, although, the theme of the article is interesting and relevant to the journal’s scope. But writing style and presentation is inadequate and inappropriate. However, I feel numerous flaws in all parts of the paper. Tile is not attractive, change it with attractive one. Improve the quality of fig 2 and 3.
A1: Thank you for the observation. We do not strive much to be attractive and catchy, just informative. We edited the manuscript based on suggestions of all three reviewers.
Comments on the Quality of English Language
Q2: The English language is not of standard quality which needs substantial improvement. The writing style is informal. Many sentences have no structure, keep consistency in writing. English language needs to be attested by a qualified English language specialist.
A2: Interesting. Please note that one of us (JGW) is a British native speaker with many many years of experience in this particular area of wheat genetics and dozens of articles behind him. If you have some specific phrasing or expression in mind, we will consider changes.
Reviewer 3 Report
The manuscript "On the possible trade-off between shoot and root biomass in wheat" by Harun Bektas, Christopher E. Hohn, Adam J Lukaszewski, J Giles Waines describes a system for comparing root biomass and aerial parts of wheat in relation to water consumption. This study is of practical importance and can be used in selection studies. However, as it is presented in the manuscript, it is not possible to evaluate this technique and the results presented.
First of all, there is no full-fledged scheme and photo of the system and plants, which does not allow us to appreciate the details. There is also no detailed description of either the number of plants studied or the dynamics, there is no description of the type of statistical methods used, software, sampling and range to assess the reliability. I failed to understand the quality of the substrate, its composition, pH, specific moisture content. A scientific publication implies the possibility of verification in any laboratory, which is impossible without specifying the data.
It is not at all clear what the lighting and temperature conditions were, how the water temperature was controlled (this greatly affects growth). It is also not clear what kind of water was used and whether fertilizers were washed away.
Also, the physiological difference between varieties is not clear - it should be discussed, are they drought tolerant, withstand flooding or something else? Again, it is not clear why there are no photographs showing the differences that the authors are talking about.
The language used in the article is somewhat alarming. For example "different wheats" sounds vague in the abstract. After all, it is not obvious what the authors mean, different types of wheat, hybrids of wheat with other types, as a consequence of distant hybridization. By the way, to what or what type do these samples belong.
In my opinion, all figures and tables require correction and addition of statistical data.
As for Figure 5, the diagram should also include an image of plants with roots. It is also not clear why the water is drawn in one part and not in accordance with the law of communicating vessels.
Figure captions should be at the bottom.
For Figures 1 and 4, I think it would be wise to highlight the area where the authors' claims of interdependence are visible.
I think this work should be expanded and accurately and qualitatively framed so that the results are clear and conclusive. After corrections, the article may be published.
If possible, authors should more clearly formulate the result of testing the hypothesis they tested in the form of an explicit conclusion, rather than reasoning.
Author Response
Responses to Reviewer 3
Q1: The manuscript "On the possible trade-off between shoot and root biomass in wheat" by Harun Bektas, Christopher E. Hohn, Adam J Lukaszewski, J Giles Waines describes a system for comparing root biomass and aerial parts of wheat in relation to water consumption. This study is of practical importance and can be used in selection studies. However, as it is presented in the manuscript, it is not possible to evaluate this technique and the results presented.
Q2: First of all, there is no full-fledged scheme and photo of the system and plants, which does not allow us to appreciate the details. There is also no detailed description of either the number of plants studied or the dynamics, there is no description of the type of statistical methods used, software, sampling and range to assess the reliability. I failed to understand the quality of the substrate, its composition, pH, specific moisture content. A scientific publication implies the possibility of verification in any laboratory, which is impossible without specifying the data.
Q3: It is not at all clear what the lighting and temperature conditions were, how the water temperature was controlled (this greatly affects growth). It is also not clear what kind of water was used and whether fertilizers were washed away.
A2-A3: Thank you. As noted above, and in the manuscript itself, here we summarize and reanalyze data collected over many years, many growing seasons, in several different experiments, with different wheats and different procedures. This is to look for some general pattern, and it appears that we found it. We do provide general descriptions of individual experimens; all details were presented in earlier articles, see references #52-57. Listing all details of many past experiments here would eat up much space and overwhelm the reader. We believe that general descriptions are sufficient for the purpose of this article.
Q4: Also, the physiological difference between varieties is not clear - it should be discussed, are they drought tolerant, withstand flooding or something else? Again, it is not clear why there are no photographs showing the differences that the authors are talking about.
A4: Please refer to Lines 331-333 and 463-466. Pavon 76 is a widely recognized wheat cultivar developed by CIMMYT, considered a “global wheat” (R. Rajaram, pers. comm) as it does well in a very wide range of environments. It has been used as a model cultivar in our previous studies. It is known for its well-documented root plasticity. Yecora Rojo is also a CIMMYT cultivar, and we have sufficient data from our team's previous studies regarding this cultivar. These two cultivars were selected based on the findings of previous research. Please see some representative studies from our team related to Pavon 76 and/or Yecora Rojo:
- Ehdaie, B., Layne, A. P., & Waines, J. G. (2012). Root system plasticity to drought influences grain yield in bread wheat. Euphytica, 186, 219-232.
- Ehdaie, B., Merhaut, D. J., Ahmadian, S., Hoops, A. C., Khuong, T., Layne, A. P., & Waines, J. G. (2010). Root system size influences water‐nutrient uptake and nitrate leaching potential in wheat. Journal of Agronomy and Crop Science, 196(6), 455-466.
- Ehdaie, B., & Waines, J. C. (2006). Determination of a chromosome segment influencing rooting ability in wheat-rye 1BS-1RS recombinant lines [Triticum aestivum L.; Secale cereale L.]. Journal of Genetics and Breeding (Italy).
- Ehdaie, B., & Waines, J. G. (2008). Larger root system increases water-nitrogen uptake and grain yield in bread wheat.
Q5: For example "different wheats" sounds vague in the abstract. After all, it is not obvious what the authors mean, different types of wheat, hybrids of wheat with other types, as a consequence of distant hybridization. By the way, to what or what type do these samples belong.
A5: Thank you Detailed descriptions of each subset of wheats tested, are listed in referenced articles. Here, for sake of brevity, we provide only a general description. If the Editor finds it advisable, we will include detailed information about all populations and sets of lines, but this would make the article excessively long.
Q6: In my opinion, all figures and tables require correction and addition of statistical data.
A6: We will be more than happy to make corrections to specific flaws, if the Reviewer lists them.
Q7: As for Figure 5, the diagram should also include an image of plants with roots. It is also not clear why the water is drawn in one part and not in accordance with the law of communicating vessels.
A7: Figure 5 aims to demonstrate how the roots follows water in the U-shaped system, illustrating the downward movement of water over time and the corresponding tracking of roots. The position of water is tracked from the transparent side rather than the plant side, allowing for adjustments in the irrigation regime.
Q8: Figure captions should be at the bottom.
A8: Fixed, thank you
Q9: For Figures 1 and 4, I think it would be wise to highlight the area where the authors' claims of interdependence are visible.
A9: We believe that the figures speak for themselves.
Q10: I think this work should be expanded and accurately and qualitatively framed so that the results are clear and conclusive. After corrections, the article may be published.
A10: Thanks, we believe that we have addressed all specific comments and suggestions of all three reviewers, improving the manuscript a great deal.
Q11: If possible, authors should more clearly formulate the result of testing the hypothesis they tested in the form of an explicit conclusion, rather than reasoning.
A11: Thank you for the comment. To us the conclusions are clear: there is a trade-off, so do not go breeding for one, as you may pay the price with the other. But it appears that different wheats may have different tipping points. We hope that this article will prompt someone to look into this issue, and the possible genetic and physiological mechanisms.
Round 2
Reviewer 3 Report
The manuscript "On the possible trade-off between shoot and root biomass in wheat" by Harun Bektas, Christopher E. Hohn, Adam J Lukaszewski, J Giles Waines describes a system for comparing root biomass and aerial parts of wheat in relation to water consumption . This study is of practical importance and can be used in selection studies. However, as it is presented in the manuscript, it is not possible to evaluate this technique and the results presented.
Some fixes have been made.
Do you want to check the spelling p ≤ 0.05 and not P = 0.05?
Why are there no statistics in table 2.
I still consider it important to expand the description of varieties.
Author Response
Responses to Reviewer 3
The manuscript "On the possible trade-off between shoot and root biomass in wheat" by Harun Bektas, Christopher E. Hohn, Adam J Lukaszewski, J Giles Waines describes a system for comparing root biomass and aerial parts of wheat in relation to water consumption . This study is of practical importance and can be used in selection studies. However, as it is presented in the manuscript, it is not possible to evaluate this technique and the results presented.
Some fixes have been made.
Q1: Do you want to check the spelling p ≤ 0.05 and not P = 0.05?
A1: Thank you, corrected.
Q2: Why are there no statistics in table 2.
A2: Table 2 is prepared to indicate the standard deviation, standard error, coefficient of variation (CV), mean, minimum and maximum values, and consequently the range values of the trials involved in each phenological development stage group. It consists of basic descriptive statistics. Since each group consists of a large number of trials, statistical analyses such as variance analysis could not be performed in this analysis. Additionally, the standard statistical data, including ANOVA, can be examined in our previous publications (see references 52-57).
Q3: I still consider it important to expand the description of varieties.
A3: We have added details of the parental lines in four mapping populations. We also added, a general description for the rest of the genotypes evaluated during the experiments. Please see lines 412-428.
These included three mapping populations consisting of doubled haploids generated from hybrids of CVs. Sonora, Foisy, and Chiddam Blanc de Mars (CBDeM) described in Hohn and Bektas [57]. Cv. Foisy was chosen by Mr. Foisy in Oregon in 1865, CBdeM originates from Ville de Paris, France, and was chosen from an English landrace. Sonora was a landrace in Durango, Mexico, known for good drought tolerance [55]. The fourth mapping population, SynOpDH, was a cross of Synthetic W7984, a synthetic amphiploid developed from the durum wheat line 'Altar 84' (Triticum turgidum L.), combined with the accession (219) 'CIGM86.940' of Ae. tauschii, and Opata M85, a well-known CIMMYT cultivar [54]. The other experiments included a variety of materials, including wild and domesticated Triticeae species with diploid, tetraploid, and hexaploid ploidy levels, Turkish landraces and cultivars, CIMMYT wheats (historical cultivars), data from allelic variation experiments with various 1RS.1BL translocation lines [55,56], and from wild and domesticated Triticeae and Aegilops sp. experiments [13,14,52,53].
The final set of data consisted of the trade-off validation experiments conducted with two semi-dwarf spring wheat cultivars from CIMMYT, Pavon 76, and Yecora Rojo (Table 1).